# “I’ve Only Just Heard About It”: Complementary and Alternative Medicine Knowledge and Educational Needs of Clinical Psychologists in Indonesia

**DOI:** 10.3390/medicina55070333

**Published:** 2019-07-03

**Authors:** Andrian Liem

**Affiliations:** School of Psychology, The University of Queensland, St. Lucia 4072, Australia; andrian.liem@uq.net.au

**Keywords:** complementary and alternative medicine, integrative medicine, knowledge, training and education, psychology, mental health, qualitative

## Abstract

*Background and objectives:* The inadequate knowledge of complementary and alternative medicine (CAM) among health professionals may put their clients at risk because clients would then find information about CAM from unreliable sources. Clinical psychologists (CPs), as health professionals, also have the opportunity to provide psychoeducation on the latest scientific CAM research for their clients. The current study aimed to explore knowledge and educational needs regarding CAM among CPs in Indonesia because previous studies on exploring CAM knowledge and educational needs regarding CAM were primarily conducted in Western countries. *Materials and Methods:* Data were collected through semi-structured face-to-face interviews with 43 CPs in public health centers (PHCs) in Indonesia. Most interviews were conducted at the PHCs where the participants worked and lasted for 55 minutes on average. The interview recordings were transcribed and were analyzed using deductive thematic analysis. *Results:* Five main themes emerged within participants’ responses regarding CAM knowledge and educational needs. First (CAM understanding), participants’ responses ranged from those with little or no prior knowledge of CAM treatments and uses, to those with much greater familiarity. Second (source of knowledge), participants’ access ranged widely in terms of references, from popular to scientific literature. Third (why is it important?), participants identified CAM as an essential part of Indonesian culture and considered it therefore crucial to have this cultural knowledge. Fourth (the challenges and what is needed?), the challenges for improving participants’ knowledge came from personal and institutional levels. Fifth (what and how to learn?), participants advised that only CAM treatments that fit in brief psychotherapy sessions should be introduced in professional training. *Conclusions:* This qualitative study discovered that CAM was neither well-known nor understood widely. Participants advised that professional associations and health institutions should work together in enhancing knowledge of CAM and incorporating CAM education into psychology education.

## 1. Introduction

The World Health Organization (WHO) defines complementary and alternative medicine (CAM) as “a broad set of healthcare practices that are not part of that country’s own tradition or conventional medicine and are not fully integrated into the dominant healthcare system. They are used interchangeably with traditional medicine in some countries” [1]. Systematic reviews of CAM efficacy and effectiveness for treatment of mental disorders have been conducted abundantly. For example, acupuncture, hypnotherapy, and meditation showed good benefit for posttraumatic stress disorder [2], and herbal medicine, particularly St. John’s wort, was reported to ameliorate mild-to-moderate depressive symptoms [3]. Moreover, the use of CAM treatments—particularly yoga and energy healing—in palliative care significantly improved the quality of life of cancer survivors [4]. In addition, yoga-based interventions are a promising non-pharmacological option for the management of depression and anxiety symptoms amongst expectant mothers [5].

Despite a growing body of scientific evidence of CAM treatments for the management of various medical conditions, systematic reviews still show methodological flaws and heterogeneity of data [6,7]. Furthermore, these reviews constrain their conclusions about the efficacy and safety of CAM treatments. Therefore, it is anticipated that health professionals would report inadequate knowledge about CAM treatments. For example, nurses in Australian hospitals reported that they had very little or no knowledge of CAM [8], and Indonesian provisional clinical psychologists reported low knowledge of CAM and predominantly relied on their friends (without psychological education background) and colleagues to learn about CAM [9]. In addition, various mental health professionals in the United States (e.g., psychologists and psychiatrists) also reported a general lack of CAM knowledge that hindered their decision in integrating CAM treatments into conventional psychotherapy [10].

The inadequate knowledge of CAM among health professionals may put their clients in a risky situation because they would then find information about CAM from unreliable sources [11]. Moreover, a survey of CAM use in 25 countries found that up to 20% of people with severe mental disorders rely only on CAM [12]. A study among American parents of children with autism spectrum disorder found that the participants did not disclose their use of CAM history with their physicians because they believed that the physician was not educated about CAM treatments [13]. Therefore, one of the recommendations in the White Paper of integrative mental healthcare is to develop comprehensive education by integrating CAM into conventional health education curricula and training [14].

CAM education and integration into the health education curriculum has been developed in health sciences since the early 2000s under various names (e.g., integrative medicine, integral medicine, and holistic health) [15,16,17]. This curriculum development particularly occurred in medical schools in the United States, where it was strongly supported by government through research and education grants. For example, more than 60% of medical schools in the United States include courses in CAM after the US government funded educational projects to accelerate the integration of CAM education in medical curricula [15,18]. The emergence of CAM integration into medical education curricula, especially in high- and middle-income countries, aligned with physicians’ and medical students’ positive attitude towards CAM education [19,20]. Similar to this finding, Australian psychologists also demonstrated their agreement towards the need for psychologists to be knowledgeable about CAM treatments [21].

A systematic review examining the effectiveness of CAM education for physicians and medical students found that CAM education not only improved their knowledge of CAM but also that participants’ communication skills and their attitudes towards CAM changed to be more positive [22]. Moreover, behavioral changes appeared in the form of more CAM referrals, and nonjudgmental discussions about CAM with clients, as well as with participants’ peers. Similar findings were also found in a retrospective evaluation survey of a unique elective course for fourth-year medical students, Humanistic Elective in alternative medicine, Activism, and Reflective Transformation (HEART) [23]. The evaluation survey was completed by 73% of 168 alumni of HEART from the 2002 to 2009 cohorts from various states in the United States. Participants perceived that the CAM knowledge that was given in the program had improved their communication skills with clients and expanded their point of view. This study showed that CAM education may enhance health professionals’ sense of humanism through exploring the preferences of the client.

The rise of CAM education integration tends to be more advanced in medical programs compared to other health programs, including psychology. Clinical psychologists (CPs), as health professionals, also have the opportunity to provide psychoeducation on the latest CAM scientific research to their clients [24,25]. CPs should be able to refer their clients to CAM practitioners, especially those who cannot use psychotic drugs or who reject conventional psychotherapies [24,26,27]. However, previous studies on exploring CAM knowledge and educational needs regarding CAM were primarily conducted in WEIRD (Western, educated, industrial, rich, and democratic) countries [28].

For instance, 9 of 22 studies in a critical review on medical students’ knowledge of CAM were conducted in the United Kingdom and United States [19]. Moreover, the amount of literature covering psychologists’ CAM knowledge and educational needs regarding CAM is very limited. Understanding these aspects is critical because it can be considered as a reference for professional psychology associations, psychology faculties, and the government to review their regulations and basic competency of CPs related to CAM, and in implementing integrative medicine services. Furthermore, recognizing the CPs’ knowledge and educational needs regarding CAM will enlighten the areas in need of attention within education curricula and ongoing professional development training of CPs.

Therefore, the current study aimed to explore knowledge and educational needs regarding CAM among CPs in Indonesia as a non-WEIRD country. Three research questions were addressed in this qualitative study: (a) How do CPs explain their knowledge of CAM?; (b) How do CPs describe, if any, their need of CAM education?; (c) How do CPs describe knowledge of CAM influences, if any, on their educational needs regarding CAM? This research may also be important for other Southeast Asian countries because of their proximity with Indonesia, in terms of culture and psychology education history [29,30,31].

## 2. Materials and Methods

### 2.1. Study Design and Procedure

A qualitative method was used because this methodology facilitates the researcher to profoundly investigate and comprehensively understand psychologists’ knowledge and the educational needs regarding CAM [32,33]. The qualitative design of this current study was constructed based on constructivist epistemology. This epistemological approach intended to explore the dynamic reality of participants’ knowledge and educational needs regarding CAM that were constructed by society [34]. There is no absolute objectivity in this epistemology because researcher’s and participant’s values and interests are entangled in the interpretation process [35].

Data were collected through semi-structured face-to-face interviews using an interview schedule that will be explained in the Instruments section. All interviews were conducted by the researcher between November 2016 and January 2017 in two areas of Special Region of Yogyakarta Province, Indonesia. Prior to this fieldwork, the researcher obtained research permission from the Indonesian Clinical Psychologist Association (IPK), which arranged for potential participants to be mailed a posted letter containing an information sheet and endorsement letter from the IPK. In this introduction letter, the researcher explained the aim of the study and asked whether the clinical psychologists were willing to be interviewed. Also, they were informed that they may choose not to participate without any consequences, however, none of them sent an opt-out email to the researcher. Before the face-to-face interview, each participant was given the chance to ask questions related to the research and asked to sign the consent form. All participants voluntarily agreed to be interviewed and were audio-recorded at their suggested time and place. They received a compensation of Rp 100,000 (equal to AUD 10).

### 2.2. Participants

A purposive sampling method for maximum variation [36] was used to select all CPs in public health centers (PHCs) in two areas of Special Region of Yogyakarta Province, Indonesia. A total of 43 participants were interviewed. Participants were aged from 25 to 42 years and they had been practicing as psychologists for between 10 months and 18 years. There was only one male participant and, therefore, the pronoun ‘she’ is used to discuss all interview responses in this thesis to maintain participants’ anonymity. Most interviews were conducted at the PHCs where participants worked, and interviews lasted for 55 minutes, on average.

### 2.3. Instruments

The interview schedule used in this qualitative study was part of a larger mixed-methods study that aimed to investigate knowledge of, beliefs and attitudes toward, experience of, and educational needs regarding CAM of CPs in Indonesia. The interview schedule was selected as a guideline (i.e., semi-structured) because of its flexibility for the interviewer to rearrange or explore beyond the interview aspects based on a participant’s responses [37]. Five aspects were explored within the interview schedule. The first aspect was knowledge of CAM; questions included how they defined CAM and how it was different from conventional medicine. The second aspect was the participant’s experience with using CAM, both in their personal and professional lives. The third aspect was spiritual–religious therapy (SRT), including the participants’ attitudes towards it and their experience with clients using it. The fourth aspect concerned CAM integration into psychological services, including regulation from government and professional associations, the best CAM integration models, and the challenge to integrate CAM into psychological services and education. The last aspect explored the participant’s educational needs regarding CAM. However, this study only discussed the findings from the first and fifth aspects. Findings from the other aspects had been reported separately [38] because of the diverse scope of analysis of these aspects. In addition, the interview schedule had been piloted, and the results of the pilot interviews had been reported elsewhere [39].

### 2.4. Data Analysis

Data analysis was initiated by transcribing the interview audio recordings. The first five audio recordings were initially transcribed by the researcher and given to the research assistant (RA) as examples of the standards for all transcripts. Then, recordings six to ten were transcribed by the RA and were evaluated for the standards assigned by the researcher before the RA transcribed the eleventh interview recording. The researcher reviewed the transcriptions of all audio recordings for accuracy by comparing the texts with the audio recording. This process also allowed the researcher to develop familiarity with the data.

Afterwards, the interview transcripts were analyzed using deductive thematic analysis due to its flexibility to both report and examine explicit and latent contents, and appropriateness in explaining specific findings in the mixed-method study [40,41]. Thematic analysis guidelines from previous studies [40,41] were followed, including (1) coding (e.g., *K* for knowledge and *E* for educational need), (2) searching for themes (reorganization of the coded data/transcripts), and (3) analysis of the reorganized data by considering the consistency of the coding, themes, and sub-themes.

### 2.5. Ethics Statement

This research was conducted in accordance with the Declaration of Helsinki and the protocol was reviewed and granted ethics approval by the Ethics Committee of the School of Psychology at the University of Queensland (16-PSYCH-PHD-08-JH) on 09/02/2016. Prior to the interviews, all participants were given a research information sheet and consent form. The consent forms were collected and safely stored in a locked filling cabinet. In the interview transcripts, participants’ personal identifiable information was removed/deidentified from transcripts to protect participants’ confidentiality.

## 3. Results

Five main themes emerged within participants’ responses regarding CAM knowledge and educational needs: (a) CAM understanding, (b) source of knowledge, (c) why is it important?, (d) the challenges and what is needed?, and (e) what and how to learn? A summary of the themes and sub-themes is presented in Table 1. Participant’s number is used in brackets to represent extracts and quotes. For example, (P6) represents a quote from Participant 6.

### 3.1. CAM Understanding

The first of the five themes was how participants understood the meaning of CAM. Responses ranged from those with little or no prior knowledge of CAM treatments and uses, to those with much greater familiarity. Also, participants defined CAM differently due to the wide range of understanding of CAM. Each sub-theme is presented in the following six paragraphs.

Almost one-third of participants expressed that they had never heard of CAM at all before, as exemplified, “I’ve only just heard about it. My colleagues were asking me about this (CAM), but I, myself, know nothing” (P6). Due to their lack of CAM knowledge, they were neither able to define nor mention an example of CAM treatment. Participants also felt unsure if some psychological techniques that they have combined into their practice were considered to be a CAM or not. However, all participants could provide examples once they knew what the term meant.

*Alternative medicine* was perceived to be more familiar than *complementary medicine* among participants. They also mentioned certain CAM treatments perceived as alternative medicine, for example, “(…) alternative medicine that uses *jamu-jamuan* (traditional herbals medicine)” (P31). Participants said that the term alternative medicine is easily found in daily life and understood because many alternative medicine practitioners advertise their services in newspapers or distribute leaflets on the street.

CAM was understood as a treatment outside conventional psychotherapy or practiced personally by clients in their homes. CAM was also defined as a psychological treatment performed by a person without a psychology education background. Participants perceived that CAM includes treatments that are not provided by a physician, as illustrated: “(…) from something to be consumed like tonic, to some actions to be done like exorcism by *dukun* (shaman)” (P12).

Participants identified CAM as a treatment combined with conventional medicine or psychotherapy, for example, pregnant clients with hypertension and anxiety used relaxation techniques in conventional psychotherapy and joined yoga to manage blood pressure and anxiety levels (P15). It was emphasized that clients usually proactively searched and used CAM based on their own beliefs and culture. Participants also explained that CPs may integrate CAM into conventional psychotherapy as a professional strategy for psychologists to be accepted by Indonesian people. For example, “We use the basic of CBT (cognitive-behavioral therapy) but inserted with religious teachings to adjust to what people understand and believe” (P30).

Three participants recognized psychological treatment, including psychotherapy and counselling, as a part of CAM. They explained that this understanding was based on the biomedicine paradigm and system in conventional health services where clients meet with a physician on the first occasion and then are referred to a CP if needed. Therefore, participants perceived that psychological treatment provided by a CP aims to complement a physician’s intervention by supporting physical healing and creating holistic health for clients.

CAM was also understood as a non (or less)-scientific treatment. The main reason given was the lack of systematic research or inadequate scientific evidence for CAM: “(…) either me who is not following the trend or what. But I feel that I haven’t found a lot research (reports) about CAM” (P4). Participants also highlighted that anyone can be a CAM practitioner without sufficient conventional medical education. In addition, there is no single guideline on how to use CAM, so every practitioner might have a different technique, compared to the strict directions for prescribing drugs among physicians. Clients’ beliefs and placebo effect were recognized by participants as two factors influencing whether CAM appears effective.

### 3.2. Source of CAM Knowledge

Based on the interviews, eight sources (sub-themes) for gaining CAM knowledge were identified (see Table 1). Participants access ranged widely in terms of references, from popular to scientific literature. Also, participants’ personal experiences of using CAM, relations with family and friends, and discussions with colleagues contributed to the enhancement of their CAM knowledge. Each source is presented in separate paragraphs below.

Participants enhanced their CAM knowledge through their colleagues, both other health professionals and CPs. For example, some gained knowledge about dietary supplements from discussions with nutritionists at the public health centers (PHCs). However, the knowledge they obtained was very basic due to time constraints for talking with other health professionals. Provisional CPs (psychologists completing a professional internship in clinical psychology under the supervision of registered clinical psychologist) who did internships with the participants could also be a source of CAM knowledge because they had more up-to-date information. Participants also gained CAM knowledge through assisting their colleagues’ research on specific CAM treatments, for example, yoga and SRT. Participants also had discussions with senior CPs and strongly believed whatever their seniors said, especially the executive members of professional associations. For example, Participant 15 exemplified, “She said, ‘Rather than listening to classical music, why we don’t listen to *gending* (traditional Javanese music instrument) or *tilawah* (Quran recitation).’ I haven’t read the journal (articles) yet. But if she spoke like that, it must have a (scientific) basis” (P15).

Family members, including partners and parents, were the most influential persons regarding development of CAM knowledge and behavior amongst participants. Parents-in-law could also be a source of CAM knowledge for participants, as described by Participant 42, “(…) my husband’s parents, in fact, prefer (to use) herbal (medicine) and my husband as well. So we also try to (use this with) our children” (P42). However, not all participants believed strongly in their family, and rather had more trust in conventional health professionals. Friends also acted as a CAM knowledge source. These friends were not from psychology nor any health education background. They could be participants’ neighbors or members of the same group activity. For example, “I know that treatment (SRT) also from friends. Sometimes we attend the same (religious) discussion forum” (P14).

Both printed and electronic mass media were used by participants to gain knowledge about CAM. Sources included magazines, newspapers, radio, and television (TV). Four participants said that they obtain basic information about *ruqyah* (exorcism in Islam) as part of SRT from religious programs on TV, for example, “I haven’t had clear (scientific) information about *ruqyah*. What I know is just from what I have seen on the television.” (P16). Participant 5 explained that the CAM knowledge she received from the radio helped her to discuss it with her clients, particularly with those from low socioeconomic status (SES) backgrounds, because they also listened to the same program.

Some participants knew particular CAM treatments from their own experience. For example, Participant 10 and Participant 15 practice yoga and meditation so they can explain to clients what they feel and the potential benefits of those treatments. In line with this, Participants 5 and 13 shared their stories with clients about herbal medicine consumption, especially during their pregnancy and when they were breastfeeding.

Numerous popular references were used by participants as their resource for CAM knowledge. For example, Participant 16 illustrated her experience with exploring dietary supplement benefits for pregnant women through popular health articles on the internet. Search engines, such as Google, were also used to find out about a particular CAM’s history or benefits. Five participants had looked for information on social media, including YouTube: “(…) now there is a lot (of instrumental music) on YouTube, sometimes I search for it there” (P22). Some popular books about CAM treatments, such as music therapy and dietary supplements, were also read to increase participants’ knowledge of CAM effectiveness.

Scientific journals and textbooks were used by some participants to find out about the effectiveness of particular CAM treatments, such as SRT, yoga, and music therapy. Two of the participants were also adjunct lecturers and read articles about CAM reviewed by their students. Participants also read journals and textbooks about CAM for their undergraduate or master theses. However, participants disclosed that sometimes they did not have time to search or could not find articles about CAM.

Participants increased their CAM knowledge by taking self-development activities, such as attending seminars, training, and workshops. They joined the programs because of their interest in the particular CAM treatment or specific developmental stage, for example, acupuncture for children with autism. Information from self-development activities were seen as more valid than popular sources or references from the internet: “(information from) browsing sometimes is not supported by robust study (…) I am confident to discuss (with clients) if have joined a seminar and obtained (more credible) information about it (CAM)” (P24).

“There was a course named Prophetic Counselling (…) in that counselling and psychotherapy, religious values were inserted” (P5). This example represented participants from Islamic universities in several provinces who obtained their knowledge about CAM, particularly SRT, in their lectures. However, some participants felt it difficult to understand the working mechanisms of SRT. By contrast, participants from a non-religious university stated that they were taught nothing about CAM in lectures. They indicated CAM was not discussed at all in the curriculum due to the lack of scientific evidence to support it. Participants also gained CAM knowledge from informal discussions or observing behaviors from their lecturers. For example, “Sometimes lecturers in the class said, ‘That’s it, give a massage here (at the palm).’ Then unconsciously, I try to apply this (acupressure) to clients” (P8).

### 3.3. Why Is It Important?

Four sub-themes, presented in the four paragraphs below, were identified when participants explained why CAM knowledge and education are important (see Table 1). Essentially, CAM was identified as part of Indonesian culture and it was therefore crucial to have this cultural knowledge. As health professionals, participants mentioned CAM knowledge and education were needed to educate clients and as part of participants’ professional development. Eventually, knowledge and education regarding CAM were needed to draft CAM integration regulations and design collaborative work with CAM practitioners in the future.

Participants perceived CAM as a part of Indonesian culture and daily life. Therefore, they commented that CPs need to have cultural sensitivity, including knowledge of CAM, so that they are able to better understand clients and provide psychoeducation about the various CAM treatments. Additionally, participants acknowledged that the conventional psychotherapy which they learnt and practiced was rooted in Western perspectives that do not always culturally fit Indonesian people’s understandings:

“Our (psychotherapy) is oriented more to the West. So, is it culturally appropriate? (It) needs modification because (we) meet with clients at PHC from diverse cultures and who use CAM (…) Consequently, CPs need cultural sensitivity like CAM understanding to understand their clients better and be able to answer questions about CAM” (P19).

Participants said that their clients still perceive CPs as having the same role as physicians. For example, in palliative care, e.g., as part of cancer treatment, clients often asked participants’ advice in choosing chemotherapy or herbal medicine. Therefore, participants emphasized that CAM knowledge and education is needed for CPs in providing information and safe recommendations or referrals for clients. Participant 28 summarized why CAM knowledge is important:

“(Clinical) recommendations are expensive. It’s OK if clients recover. But, if their condition becomes worse, then it is dangerous (…) when a client asks about CAM, the CP should be able to answer even if only with a short explanation. If the CP’s response is only ‘I do not know’ then it can lessen the CP’s credibility and trust from the client. Also, it will increase the risk of the client looking up information from invalid (not credible) sources which, in the end, endangers the client” (P28).

Participants explained that their clients came from various backgrounds and with numerous problems. Therefore, they need to keep up-to-date information and upgrade their knowledge and skills, including about CAM. Thus, knowing and learning about CAM could be part of lifelong learning, and CPs might specialize in one particular CAM treatment. In addition, participants suggested that CAM education undertaken by CPs could be used as one requirement to renew their registration. They also highlighted that CPs need to be certified before integrating CAM into their practice. Participants suggested that their profession might learn from other health professions that already integrate CAM into their practice, for example, “Like ob-gyns (obstetrics and gynecologists), they can do hypnobirthing. That will make their services more optimal” (P39).

Participants mentioned that regulation is strongly needed if CAM is to be integrated into psychological services. However, CPs must have the knowledge of CAM prior to formulating such regulations. Therefore, CAM education is strongly recommended for CPs since, “Not all CPs have (CAM) knowledge or have been taught about CAM so there is a need for CAM education” (P38). Apart from regulation, CAM knowledge and education are essential for participants when planning collaboration with CAM practitioners.

### 3.4. The Challenges and What Is Needed?

Four sub-themes relating to challenges were identified from the interviews, which are presented separately in the four paragraphs below. The challenges for improving participants’ knowledge came from personal and institutional levels. Financial and time constraint issues hindered participants from participating in CAM workshops. Additionally, participants expressed that the content of CAM seminars was too basic and did not meet their needs in clinical practice. Approval from participants’ institutions to attend CAM self-development activities was another challenge faced.

Participants criticized the expensive cost to join CAM seminars, training, or workshops to improve their knowledge and skills. Participants’ income as a CP at a PHC, which is lower than other health professionals, is insufficient to cover the registration fees. Additionally, participants regretted that some high-cost training did not make them certified to do particular treatments and, hence, they were not cost-effective.

Working at a PHC demands time and energy, said participants. Consequently, it is difficult for participants to increase their knowledge by regularly joining CAM self-development activities. When a particular training event is held in a different city and on a weekend, it is still hard to take part if they have a child. Moreover, searching for and reading the latest journal articles were considered time-consuming for some participants, so they would just practice what they learnt from university.

Participants underlined that it is a challenge to find a trainer who is an expert in CAM as well as understands psychology. Most CAM training facilitators were from non-psychology backgrounds, so what they explained was too basic or irrelevant from what participants needed. Also, it is important to know the competency of the educator in CAM, as Participant 12 criticized, “Who is assessing their competence?” (P12). Furthermore, finding credible CAM education institutions that are recognized by other health professionals is another issue faced. As an alternative solution, participants suggested that professional associations might organize CAM training so that it is perceived as more credible.

Participants explained that it is difficult to ask permission from their institutions to attend CAM training and workshops that are held during the work week. Participants also perceived that their institutions, PHCs, and the Health Department show less support by offering insufficient training and funding. However, some institutions supported participants through providing facilities, for instance, a computer and internet connection in the participant’s room so they could search for CAM references in a timely manner.

### 3.5. What and How to Learn?

This last theme presented some principles of CAM education needed by participants and is merged into six sub-themes (Table 1) which are discussed in the following six paragraphs. Participants advised that only CAM treatments that fit in brief psychotherapy sessions should be introduced in professional training. Since CAM was perceived as part of Indonesian culture, participants suggested CPs should know about the CAM treatments commonly used in daily life. Regarding their professionalism, participants emphasized that only less harmful and scientific-based CAM treatments should be included in psychology education. Lastly, participants recommended comprehensive content and experiential learning strategies when teaching CAM for CPs.

Due to time constraints and pressures at PHCs, participants agreed that treatments should be made easy to conduct during brief sessions and should be introduced as part of psychology education. For example, music therapy should be taught because it can be used anywhere, as should acupressure because of its simplicity. It was found that neuro-linguistic programming (NLP) was the most commonly endorsed treatment, followed by hypnotherapy, brain gym, and eye movement desensitization and reprocessing (EMDR). Those treatments were favored because they produce quick effects and clients like them.

Participants explained that stigma towards mental disorders still occurs in Indonesia and it discourages people from visiting psychology services. Therefore, participants suggested that CPs should learn and use treatments that become part of clients’ everyday lives, for instance, massage therapy and herbal medicine. By integrating these treatments, participants indicated that this would encourage people to come to CPs and help reduce the stigma. Participants agreed that SRT should be taught in professional programs because spirituality and religion are very important to Indonesian people’s lives: “Our people are relatively religious, whatever their religion. Thus, therapies with spirituality and religious influences need to be taught” (P34).

Participants showed their concern regarding clients’ safety when using CAM in psychological practice. Therefore, they suggested CPs should only learn less harmful and non-instrumental treatments such as acupressure and meditation. Acupuncture and herbal medicine were the least recommended treatments because participants were not taught about them at university and they did not have a license to practice them. However, participants suggested that CPs need to know about herbal medicine, as exemplified: “CPs also need to know about pharmacology, and herbal medicine can be inserted into that topic within lectures” (P12).

Mind–body treatments, such as yoga and meditation, were proposed by participants to be included in psychology education because of the connection between physical and psychological conditions. However, not all Islamic universities might want to teach it: “But about yoga and meditation, Islamic universities tend to reject (to teach them) because they’re basically from either Buddhism or Hinduism” (P20). Participants also highlighted that the CAM treatments taught in psychology programs have to be scientifically based. Therefore, some participants doubted hypnotherapy would be taught, as illustrated, “not all CPs favor hypnotherapy because the scientific evidence is still weak” (P23).

Participants disclosed that information about CAM is rare in psychology programs and undervalued compared to conventional psychotherapy. Therefore, participants commented that it is important that every aspect of CAM be taught because this will make them more confident and understand CAM more comprehensively. Basic information, such as CAM philosophy and history, is essential in order to find common ground and avoid dissenting opinions among CPs. However, some participants also emphasized that practical information, such as working mechanisms and side effects, also needs to be known.

Role-plays and visits to CAM practitioners or inviting CAM practitioners to the classroom were the most recommended ways for participants to learn about CAM. These hands-on experiences and learning strategies may enhance their knowledge and skills with CAM. Participants also suggested that the educators should be able to teach CAM based on what CPs need in their practices, for example, by using case studies.

## 4. Discussion

This current qualitative study aimed to explore Indonesian CPs’ knowledge and educational needs regarding CAM. Lack of CAM knowledge among participants in this study supports a similar finding which was reported by Australian psychologists [42]. CAM, particularly complementary medicine, is an unfamiliar term and was interpreted with various meanings. In line with previous studies in the psychology community in Indonesia [43] and psychologists in Australia [44], participants in this qualitative phase also perceived that CAM has inadequate scientific evidence to support it. 

Time constraints and pressures in PHC, as discussed by participants, might explain why only a few of the participants read scientific journals as their references for CAM. However, the difficulty in finding research published on CAM in psychology journals might illustrate that the mainstream psychology journals have a low acceptance rate for CAM studies, as was found in previous studies [24,44,45]. This phase also discovered that the actions and statements by lecturers and executive members of professional associations may also be used as references, even when not supported by adequate scientific evidence.

The interviews revealed three reasons for the importance of CAM knowledge and education for participants. First, CAM was perceived as part of clients’ culture, and participants often received questions from clients related to CAM treatments. Therefore, cultural sensitivity (i.e., non-prejudiced attitude towards CAM treatments) to be able to answer clients’ questions might be achieved by learning more about CAM. Second, CAM knowledge was needed to provide psychoeducation on the latest CAM scientific research to their clients as suggested in a previous study [24]. Moreover, accurate information about CAM is essential to protect clients from the malpractice of CAM practitioners as also mentioned by Australian psychologists [44]. Lastly, CAM education was needed in order to formulate the regulation of CAM integration into psychological services, a view which was also found in previous studies among health professionals in Australia and Bahrain [46,47].

Participants mentioned several challenges in increasing their knowledge about CAM. The main barriers to joining self-development activities, such as training and workshops, was that they were financially expensive and participants had little time to do so. Another challenge was related to the relatively low income in PHCs that was not enough for participants to update their knowledge and skills about CAM. Some PHCs provide financial support for participants to join self-development activities, but in limited amounts. Therefore, participants expected their institution to raise the budget and professional associations to organize inexpensive CAM training and workshops. Working pressure in PHCs also inhibited participants from upgrading their knowledge about CAM because of the difficulty in getting permission from the institution to attend such workshops. Professional associations were expected to supervise the credibility of CAM education institutions and also the trainers. This finding aligned with psychologists in Australia who also emphasized the importance of professional associations’ surveillance of CAM training and practice [42]. Participants expected that professional associations would acknowledge the credits from CAM workshops as one component of psychology practice license renewal.

There was dissenting opinion among participants about which CAM treatments should be learnt by CPs. Based on efficiencies demanded by PHCs, numerous participants recommended NLP be inserted into psychology education so they could use it in their practice. Conversely, some participants discouraged CAM treatments with insufficient scientific evidence, including NLP, from being taught by psychology faculties. Mind–body treatments, such as meditation, were perceived as a safe CAM treatment that CPs may learn and practice. This finding supported the previous study that found that psychologists in Australia showed positive attitudes towards meditation integration into conventional psychotherapy [44]. However, in this qualitative phase, it was also explored that Islamic universities might not teach meditation and yoga because these treatments are rooted in different religions’ teachings. As part of Indonesian peoples’ everyday lives, participants suggested that SRT should be taught in psychology education. This supports previous research where physicians in Indonesia also showed positive acceptance towards SRT in conventional medicine [48].

Despite the fact that several procedures were taken to support the trustworthiness of this qualitative study, there were three limitations that need to be considered. First, the sex proportion was imbalanced, and only one male participant was interviewed. The analysis might not accurately represent male clinical psychologists’ knowledge and educational needs regarding CAM. Therefore, the future study may investigate knowledge and educational needs regarding CAM among males CPs. Second, all interviews were conducted individually, and this was a disadvantage when a participant shared very limited responses due to his/her insufficient knowledge on the topic. Hence, individual interviews can be enriched by combining with focus group interviews where additional layers of data may be obtained from interaction between participants. Third, psychological education in Indonesia may share similarity with other Southeast Asian nations, however, it may not be entirely possible to generalize the findings from Indonesian CPs into different cultural settings. Therefore, future study may be conducted cross-culturally with other psychologists from different nations as participants to allow comparison with the current findings.

## 5. Conclusions

This qualitative study with 43 clinical psychologists in Indonesia discovered that CAM was neither well-known nor understood widely among participants. Participants explained reasons why CAM knowledge and education were important to them. For example, having cultural sensitivity and including knowledge of CAM is essential to better understanding clients. Therefore, participants advised that professional associations and health institutions should work together in enhancing knowledge of CAM and incorporating CAM education into psychology education.

## Figures and Tables

**Table 1 medicina-55-00333-t001:** Themes and sub-themes for CAM knowledge and educational needs.

Theme	Sub-Theme
CAM understanding	“I’ve only just heard about it”
	Familiarity
	Outside of conventional medicine or psychotherapy
	Companion of conventional medicine or psychotherapy
	Psychological treatment considered as CAM
	Less scientific
Source of knowledge	Colleagues
	Family and friends
	Mass media
	Personal experience
	Popular reference
	Scientific reference
	Self-development activities
	University
Why is it important?	CAM is part of Indonesian culture
	Provide information and recommendation or referral
	Lifelong learning and certification
	Formulate CAM regulation and collaboration with CAM practitioners
The challenges and what is needed?	Costly registration fees
	Time constraints
	Credible institutions and educators
	Support from institutions
What and how to learn?	Brief and easy treatments
	Harmless and non-instrumental treatments
	Daily life and less stigmatized treatments
	Mind–body and scientific-based treatments
	Comprehensive understanding
	Experiential learning process

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
