# Peer review of "“I’ve Only Just Heard About It”: Complementary and Alternative Medicine Knowledge and Educational Needs of Clinical Psychologists in Indonesia"

_medicina, 2019, doi:10.3390/medicina55070333_

Reviewer 1 Report

Although the topic is clinically relevant, the manuscript does not provide new insights into complementary and alternative healing practices, and I found several areas of the article underdeveloped.

Specific comments:

- Please change "showed that these treatments significantly improved the quality of life of cancer survivors" to "significantly improved the quality of life of cancer survivors".

- Specifically, yoga-based interventions have also shown benefit for the management of depression and anxiety symptoms amongst expectant mothers (citation: ncbi.nlm.nih.gov/pubmed/30712750).

- Words like evidence, knowledge, information and research occur commonly in academic writing. These words never take a plural s. They are always singular and the verb is therefore always singular too.

- Please temper the statement "Despite these plentiful scientific evidences..." Despite a growing body of scientific evidence of CAM therapies for the management of various medical conditions, systematic reviews still show methodological flaws and often highlight heterogeneity of data, limiting definite conclusions about their efficacy and safety (citation: ncbi.nlm.nih.gov/pubmed/19327658; and ncbi.nlm.nih.gov/pubmed/29197739).

- In the introduction, it is important to highlight that CAM also includes the use of herbal supplements. The authors have only mentioned the use of nonpharmacologic interventions, with no mention of herbal remedies. In particular, St. John's wort remains among the top-selling botanical products in the United States and many brands are now available and sold over the counter as dietary supplements, with purported effectiveness for mild-to-moderate depression (citation: ncbi.nlm.nih.gov/pubmed/28064110).

- Please change "how do" to "how does".

- Please change "proximity of culture and psychology education history with Indonesia" to "proximity in terms of culture and psychology education history with Indonesia".

- Please rephrase "The qualitative design of this current study was constructed based on constructivist epistemology that intended to explore the dynamic reality that constructed by society". This is a very long and convoluted sentence.

- How was the sample size determined? Did the author conduct any pilot testing?

- Please rephrase "This current qualitative study aimed to explore CPs’ knowledge and educational needs of CAM as a part of mixed-method research on knowledge of, beliefs and attitudes toward, experience of, and educational needs regarding CAM of CPs in Indonesia". This is a very confusing and convoluted sentence.

- What is meant by "cultural sensitivity" in the context of CAM in Indonesia? Please give a concrete example to illustrate this.

- The overarching challenge of CAM is overlooked in this article. Today, there is still a paucity of clear and consistent evidence and no scientific consensus to inform modern CAM practice. This translates into poor knowledge or uncertainty amongst healthcare practitioners and the wider public health workforce.

- For future work, focus group interviews with select participants could be carried out, and more detailed thematic analyses would have enhanced the present study.

Author Response

1. Although the topic is clinically relevant, the manuscript does not provide new insights into complementary and alternative healing practices, and I found several areas of the article underdeveloped.

We thank Reviewer 1 for their time to review our manuscript and the constructive feedback. As represented in the title, our manuscript focused on clinical psychologists’ knowledge and educational need of complementary and alternative medicine (CAM). We have explained in the manuscript that inadequate knowledge of CAM among health professionals, including clinical psychologists, may put their clients in risky situation because they then would find information about CAM from unreliable sources (L58-59). Additionally, the amount of literature covering clinical psychologists’ CAM knowledge and educational needs of CAM is very limited (L99-100). Therefore, the novelty and importance of this manuscript are positioned on CAM education for psychologists in particular and mental health professionals in general.

2. Please change "showed that these treatments significantly improved the quality of life of cancer survivors" to "significantly improved the quality of life of cancer survivors".

 The sentence has been revised (L44).

 3. Specifically, yoga-based interventions have also shown benefit for the management of depression and anxiety symptoms amongst expectant mothers (citation: ncbi.nlm.nih.gov/pubmed/30712750).

 We thank the Reviewer for this suggestion. The sentence and citation have been inserted to the manuscript (L44-46).

4. Words like evidence, knowledge, information and research occur commonly in academic writing. These words never take a plural s. They are always singular and the verb is therefore always singular too.

             These words have been checked to be written in the singular form (i.e. ‘evidence’ L47).

5. Please temper the statement "Despite these plentiful scientific evidences..." Despite a growing body of scientific evidence of CAM therapies for the management of various medical conditions, systematic reviews still show methodological flaws and often highlight heterogeneity of data, limiting definite conclusions about their efficacy and safety (citation: ncbi.nlm.nih.gov/pubmed/19327658; and ncbi.nlm.nih.gov/pubmed/29197739).

 The statement has been amended. One of the recommended references and additional publication (the most up to date) have been incorporated into the manuscript (L47-50).

6. In the introduction, it is important to highlight that CAM also includes the use of herbal supplements. The authors have only mentioned the use of nonpharmacologic interventions, with no mention of herbal remedies. In particular, St. John's wort remains among the top-selling botanical products in the United States and many brands are now available and sold over the counter as dietary supplements, with purported effectiveness for mild-to moderate depression (citation: ncbi.nlm.nih.gov/pubmed/28064110).

 Herbal medicine and the recommended reference have been mentioned in the Introduction section (L41-43).

 7. Please change "how do" to "how does".

 The subject of this sentence are clinical psychologist (CPs - plural), so we did not change ‘How do’ as suggested by Reviewer 1.

 8. Please change "proximity of culture and psychology education history with Indonesia" to "proximity in terms of culture and psychology education history with Indonesia".

 The sentence has been changed (L111-112).

 9. Please rephrase "The qualitative design of this current study was constructed based on constructivist epistemology that intended to explore the dynamic reality that constructed by society". This is a very long and convoluted sentence.

 The sentence has been clarified into: “The qualitative design of this current study was constructed based on constructivist epistemology. This epistemological approach intended to explore the dynamic reality of participants’ knowledge and educational needs of CAM that constructed by society.” (L117-119).

 10. How was the sample size determined? Did the author conduct any pilot testing?

 The sample size in this qualitative study was determined with a purposive sampling method for maximum variation (as already mentioned in L136). We conducted a pilot testing (pilot interviews) of interview schedule and the results had been reported elsewhere (this information has been mentioned in L158-160).

 11. Please rephrase "This current qualitative study aimed to explore CPs’ knowledge and educational needs of CAM as a part of mixed-method research on knowledge of, beliefs and attitudes toward, experience of, and educational needs regarding CAM of CPs in Indonesia". This is a very confusing and convoluted sentence.

 The sentence has been clarified into: “This current qualitative study aimed to explore Indonesian CPs’ knowledge and educational needs of CAM.” (L432-433)

 12. What is meant by "cultural sensitivity" in the context of CAM in Indonesia? Please give a concrete example to illustrate this.

 Cultural sensitivity in the context of CAM can be demonstrated by non-prejudiced attitude towards CAM treatments, particularly for CAM treatment that commonly used by Indonesian people. This clarification has been incorporated to the manuscript (L447-448).

 13. The overarching challenge of CAM is overlooked in this article. Today, there is still a paucity of clear and consistent evidence and no scientific consensus to inform modern CAM practice. This translates into poor knowledge or uncertainty amongst healthcare practitioners and the wider public health workforce.

 We thank the Reviewer for this comment. Our qualitative findings clearly support Reviewer’s statement ‘… poor knowledge or uncertainty amongst healthcare practitioners …’ because participants (clinical psychologists) admitted that they had insufficient knowledge of CAM, unfamiliar with CAM, and perceived that CAM treatments have inadequate scientific evidence to support it (L434-437). We also agree that ‘…a paucity of clear and consistent evidence and no scientific consensus to inform modern CAM practice’ are one of the biggest challenges in integrating CAM into conventional medicine practice. However, we would argue this challenge may more relatable to systematic review study. Meanwhile, in our qualitative study, the findings (thematic analysis) were driven by interviews results with participants who had lack of CAM knowledge. Regardless of this difference of opinions, the difficulty experienced by participants in finding research published on CAM in psychology journals (L441-443) may reflect Reviewer’s concern about inadequate scientific evidence to support CAM.

 14. For future work, focus group interviews with select participants could be carried out, and more detailed thematic analyses would have enhanced the present study.

 We have added focus group interviews as one of the recommendations for future study (L486-490). However, we would like to argue that five main themes emerged from our thematic analysis have adequately portrayed participants’ responses on CAM knowledge and educational need for CAM. More detailed analyses, as suggested by the Reviewer, might more appropriate for different qualitative technique such as content analysis.

Reviewer 2 Report

This qualitative study focused on 43 clinical psychologists, which researched the knowledge and educational needs of complementary and alternative medicine (CAM) on their practice in Indonesia. Data were collected through semi-structured face-to-face interviews in Indonesia public health centers, using deductive thematic analysis. Results were according to 5 main themes: CAM understanding, source of knowledge, why is it important, the challenges and what is needed, and what and how to learn. The author concluded that professional associations and health institutions should work together in enhancing knowledge of CAM and incorporating CAM education into psychology education.

The qualitative paper presented the knowledge and educational needs of CAM among the Indonesian clinical psychologists. It may be useful in Indonesia or even other area about CAM. The manuscript was well done and readable, however, in addition to time limitations (lines 436-442), I suggest to add a section of limitation about the study design and semi-structure interview in the “discussion” will be better.

Author Response

1. The qualitative paper presented the knowledge and educational needs of CAM among the Indonesian clinical psychologists. It may be useful in Indonesia or even other area about CAM. The manuscript was well done and readable.

 We thank the Reviewer for this positive feedback.

 2. However, in addition to time limitations (lines 436-442), I suggest to add a section of limitation about the study design and semi-structure interview in the “Discussion”.

 The Reviewer might misread the paragraph since ‘time limitations’ in these lines referred to participants’ explanation why only a few of them accessed/read scientific journals as their references for CAM. We have changed ‘time limitations’ to ‘time constrains’ to improve its clarity (L438). We have also added additional limitation related to the study design (individual interviews) at the end of Discussion section to accommodate Reviewer’s suggestion about study design (L486-490).

Round  2

Reviewer 1 Report

Thank you for the revisions.

- Please note spelling error. It should be "Time constraints".